# The Efficacy and Safety of Simultaneous Vaccination with Polysaccharide Conjugate Vaccines Against Pneumococcal (13-Valent Vaccine) and *Haemophilus influenzae* Type b Infections in Children with Juvenile Idiopathic Arthritis Without Systemic Manifestations: A Prospective Cohort Study

**DOI:** 10.3390/vaccines13020177

**Published:** 2025-02-12

**Authors:** Ekaterina Alexeeva, Tatyana Dvoryakovskaya, Anna Fetisova, Ivan Kriulin, Elizaveta Krekhova, Anna Kabanova, Vladimir Labinov, Elizaveta Labinova, Mikhail Kostik

**Affiliations:** 1Department of Pediatric Rheumatology, National Medical Research Center of Children’s Health, Moscow 119991, Russia; alekatya@yandex.ru (E.A.); tbzarova@mail.ru (T.D.); anna_534@mail.ru (A.F.); 79671819676@yandex.ru (I.K.); lizakrek@mail.ru (E.K.); anyyyta96@yandex.ru (A.K.); 2Clinical Institute of Children’s Health Named After N.F. Filatov, Department of Pediatrics and Pediatric Rheumatology, I.M. Sechenov First Moscow State Medical University (Sechenov University), Moscow 119991, Russiaelizaveta.labinova@yandex.ru (E.L.); 3Hospital Pediatry, Saint-Petersburg State Pediatric Medical University, Saint-Petersburg 194100, Russia

**Keywords:** juvenile idiopathic arthritis, JIA, vaccines, pneumococcal vaccine, *Haemophilus influanzae* type b vaccine, methotrexate, biologic treatment

## Abstract

**Background**: Immunosuppressive therapy (methotrexate and biological agents) for juvenile idiopathic arthritis (JIA) is associated with an increased risk of severe infections, higher infection rates, treatment interruptions, failure to achieve disease remission, and recurrent disease flares. Our study aimed to evaluate the safety and efficacy of simultaneous immunization with 13-valent polysaccharide conjugate vaccines (PCV13) against *S. pneumoniae* (SP) and *Hemophilus influanzae* type b infections (HibV) in children with JIA without systemic manifestations. **Methods**: A total of 371 non-systemic JIA patients who received 13PCV and HibV were included in this prospective cohort study. In every patient, we evaluated clinical, laboratory, anti-SP, and anti-Hib IgG antibodies before vaccination, three weeks after, and six months after, and all adverse events (AEs) were collected during the study. The number and duration of acute respiratory infection (ARI) episodes and requirements for antibacterial treatment and AE six months before and after the baseline were collected. **Results**: The levels of the Ig G anti-SP and anti-Hib antibodies increased in the 3 weeks after vaccination; then, anti-SP antibodies slightly decreased and anti-Hib antibodies remained increased during the whole study, as well as in a part of the patients with a protective titer. During the study, there were no patients with significant flares, and the main JIA outcomes gradually decreased during the trial. The number of patients with uveitis remained equal, as well as the part of the patients with active, low-active, and inactive uveitis. There was no significant rise in the hs-CRP or S100 protein after the vaccination. Previous or ongoing treatment with non-biological (*p* = 0.072) and biological (*p* = 0.019) disease-modified anti-rheumatic drugs affected the Hib and did not affect the anti-SP protective titer at the end of the study. Within 6 months following vaccination, the number of ARI episodes (*p* < 0.001) and the number of courses of antibacterial treatment (*p* < 0.0001) decreased twice. The median duration of ARI episodes decreased four times (*p* < 0.0001). Mild AEs (injection site reactions and short-term fever episodes) were found in 58 (15.6%) patients with JIA, and 1 patient (0.2%) developed an SAE. **Conclusions**: Simultaneous vaccination against pneumococcal and Hib infections reduces the frequency and duration of episodes of ARI, as well as the number of courses of antibacterial drugs, and does not lead to significant JIA flares. The number of reported AEs is consistent with what was expected.

## 1. Introduction

Juvenile idiopathic arthritis (JIA) is the most common chronic rheumatic disease of unknown etiology in children with serious short- and long-term disabilities [1,2]. The implementation of biological drugs into the treatment protocols of JIA has changed the prognosis of the disease. The high efficiency of biological therapy ensures a high quality of life for children and their families and the psychological, social, and professional adaptation of patients with JIA [3]. Biologic and non-biologic anti-rheumatic drugs are the main treatment approaches for JIA and might be complicated by infections, predominantly upper respiratory tract diseases, otitis media, and sinusitis [4,5]. *Pneumococcus*, *influenza*, varicella-zoster virus, and measles are the most frequent infectious diseases in immune-compromised patients, with more severe infectious disease courses and requiring intravenous antibiotics and immunoglobulin. Infectious episodes are the main reasons for interrupting current therapy, disease flares, and the achievement of remission failure [6,7,8]. According to WHO estimates, *Streptococcus pneumoniae* causes the death of about half a million children under the age of 5 worldwide every year, and most of these deaths occur in developing countries [9]. Until 2000, the number of cases of *Haemophilus* infection was 8.13 million cases in children aged 1–59 months, among which the infection ended fatally in 371 thousand children [10]. *P. aeruginosa*, *H. influenzae*, *M. catarrhalis*, and their associations were statistically more common causes of pneumonia in adult patients with rheumatoid arthritis (RA) than in patients without rheumatoid arthritis. It was found that RA is an independent risk factor for mortality from pneumonia [11]. The main method of preventing these infections is vaccination. Vaccination of children with rheumatic diseases is a difficult task for pediatric rheumatologists, especially for vaccines not included in the calendar of preventive vaccinations. Non-live vaccines are recommended for all children with rheumatic diseases, regardless of the therapy [12]. Incomplete vaccination is typical for many countries. In Slovenia, 35% of patients with rheumatic diseases had incomplete vaccinations, and the same in Russia [12,13]. Moreover, vaccination coverage decreased during the COVID-19 outbreak [14]. The protective level of vaccine-specific antibodies might be lower in patients with immune-mediated diseases predominantly due to immunosuppressive treatment [11,15]. Many national and international professional associations encourage doctors and patients to be vaccinated [11]. Fear of a flare of the disease or low efficacy of the vaccination are the main factors against vaccination in parents and physicians [13,16,17]. Vaccination coverage in children with JIA in the last decades in the Russian Federation against the major vaccine-controlled infections amounts to 50–58% and remains low [16].

Our study aimed to evaluate the efficacy and safety of the simultaneous immunization with polysaccharide conjugate vaccines against pneumococcal (13-valent vaccine) and *Haemophilus influenzae* type b (Hib) infections in children with JIA without systemic manifestations.

## 2. Methods

### 2.1. Study Design and Patient Selection

We included 371 JIA patients who met the inclusion/exclusion criteria in a prospective cohort study with retrospective parts. The selection of the patients were conducted voluntary. In all cases the legal representatives and the patients older than 15 years confirmed their agreement in the study. The study was carried out in pediatric rheumatology departments of the National Medical Research Center of Children’s Health, Moscow, Russian Federation, and Saint-Petersburg State Pediatric Medical University, Saint-Petersburg, Russian Federation, in 2018–2020. All patients had daily medical observations during baseline (Day 0, D0) and three weeks later (Days 1–22, D1–D22). The end of the study (EOS) observation was conducted six months after the baseline (Figure 1).


*
**Inclusion criteria**
*


-Diagnosis of JIA according to the ILAR (International League of Associations for Rheumatology) criteria [18];-Non-systemic JIA categories: oligoarthritis (persistent and extended), polyarthritis (rheumatoid factor positive and negative), enthesitis-related arthritis, and psoriatic arthritis;-Age at the study inclusion 2–18 years;-Treatment with either non-biologic or biologic disease-modified anti-rheumatic drugs (DMARDs) or their combination;-No previous vaccination against pneumococcal and Hemophilus influenza type b infection, except scheduled vaccination of this infection in the first year of life according to the national vaccine schedule [18].


*
**Exclusion criteria:**
*
-Any information on vaccine intolerance in the past;-JIA with systemic onset or undifferentiated arthritis;-Patients with ongoing acute infection illness or 4 weeks before the baseline.


### 2.2. Vaccination

We used polysaccharide conjugate vaccines against pneumococcal (13-valent vaccine, Pfizer, Dublin, Ireland) and *Haemophilus influenza* type b infection (GlaxoSmithKline, Rixensart, Belgium). Both vaccines (0.5 mL) were injected one by one at a 10 min interval subcutaneously in the left (*Haemophilus influenza* type b vaccine) and right (pneumococcal vaccine) deltoid areas. Every patient was under the physician’s close monitoring for at least 30 min after the last vaccine injection.

### 2.3. The Assessments of the Study


(**i**)
**Demography**



Sex, onset age of JIA, duration of JIA before the baseline, and ongoing treatment.


(**ii**)
**Clinical assessment of JIA**



Number of active joints and clinical assessment of JIA activity with visual analogue scale (VAS): physician’s VAS (MD VAS) and patient/parental VAS (pVAS) and Childhood Health Assessment Questionnaire (CHAQ) [19]. The remission of JIA was determined according to the criteria of C. Wallace [20]. Remission on medication is mentioned if the patients had all the abovementioned criteria of remission during a month without flares.


(**iii**)
** Laboratory assessment of JIA activity**



In every patient, we evaluated routine testing: complete blood cells and the erythrocyte sedimentation rate (ESR). We also evaluated specific tests: high-sensitivity CRP (hs-CRP) and calprotectin (S-100) (MRP8/14 complex).

We employed commercial test kits to assess the concentration of high-sensitivity C-reactive protein (hsCRP) (Biomerica, Hamburg, Germany), employing a solid-phase indirect enzyme immunoassay and kits for measuring the levels of MRP8/14 (S100A8/A9, calprotectin) (Bühlmann, Schönenbuch, Switzerland) for conducting a sandwich ELISA. The preparation of blood samples for testing, as well as the registration of ELISA results, were carried out by the protocol described above for assessing the immunogenicity of vaccines. Threshold values for the hsCRP (upper limit 8.2 mg/L; analytical sensitivity 0.1 mg/L) and S-100 protein (upper limit 2.0 μg/mL; analytical sensitivity 0.4 μg/mL) were used to determine JIA activity.


(**iv**)
**Assessment of the vaccine-specific antibodies**



Venous blood samples were taken on the day of vaccination, 3 weeks after it, and 6 months (acceptable range up to 3 days) after it. Blood in a volume of 5 mL was placed in tubes with EDTA and kept at room temperature for 30 min in a vertical position. The tubes were then centrifuged at 2000 rpm for 10 min. Plasma samples were placed in Eppendorf bags (Deltalab S.L, Barcelona, Spain) and frozen at −80 °C (ULT Freezer, Haier, Qingdao, China) for subsequent storage. Before the ELISA, the plasma samples were thawed at room temperature, mixed for several seconds on a Vortex, and centrifuged at 2000 rpm for 10 min. Plasma samples at a dilution of 1:101 were used for the test.

Anti-SP (*S. pneumonia)* titers (total IgG titer to serotypes 1–5, 6B, 7F, 8, 9N, 9V, 10A, 11A, 12F, 14, 15B, 17F, 18C, 19A, 19F, 20, 22F, 23F, and 33F) and anti-Hib in plasma samples were determined using commercial EIA PCP IgG (Test Line Clinical Diagnostics, Brno-Královo Pole, Czech Republic) and HibIgG ELISA (IBL International Gmbh, Hamburg, Germany) kits, respectively. The lower sensitivity threshold of the ELISA for anti-SP (according to the manufacturer’s instructions) was 2 U/mL and the upper sensitivity threshold was 270 U/mL; for anti-Hib, it was −0.07 and 4 µg/mL, respectively.

For titer values below and above the specified ELISA sensitivity values, a threshold value was indicated. Coefficients of variation (according to the manufacturers’ instructions) for the EIA PCP IgG test were 3.8% (intra-assay) and 8.1% (inter-assay), and for the HibIgG ELISA were ≤10% and 9–12%, respectively. Registration of the ELISA results (staining intensity of the contents of the wells of a 96-well plate) was carried out using an Infinite 200 M plate reader (Tecan, Grödig, Austria) at 450 nm.

We assessed all the abovementioned measures at baseline (day of vaccination), three weeks after baseline, and six months after the baseline.


(**v**)
**The safety of vaccination**



All adverse (vaccine-associated, disease-associated, and common) events were recorded (AE) during the six-month follow-up period after the baseline.

(a)Vaccine-associated AE included any acute reactions (first 30 min after injections) and delayed reactions (fever, malaise, fatigue, and injection-site reactions).(b)Disease-associated AE mentioned any clinical and/or laboratory signs of JIA flare or that required any changes in the treatment.(c)Common AEs included any events that occurred in the six-month follow-up period, except the abovementioned AEs.

The data about acute AEs were collected during the baseline period, and all remaining AE data were collected at the three-week and six-month observation visits by asking the patients and parents and analyzing the medical charts of the patients in a retrospective manner.

(**vi**)
**Respiratory infection episode analysis**


The number and duration of all respiratory illness episodes, irrespective of etiologic factors, and requirements for antibiotic treatment six months before the baseline were compared with six months after the baseline by asking the patients and parents and analyzing the medical charts of the patients on a retrospective basis.

### 2.4. The Study Outcomes

The main study outcomes were as follows:

(***i***)
*
**Analysis of seroconversion**
*


The effectiveness of vaccines was assessed by their immunogenicity, namely by the proportion of patients in whom a protective antibody titer was determined 3 weeks and 6 months after vaccination and/or in combination with an increase in the titer ≥ 2 times from the initial one.

An antibody titer (IgG) to Streptococcus pneumoniae (anti-SP) ≥ 7 U/mL and to *Haemophilus influenzae* type b ≥ 1.07 µg/mL (equivalent to U/mL) was considered protective.

(***ii***)***Assessment of JIA-related outcomes during this study:*** active joints, joints with limited motion, morning stiffness duration, MDVAS, pVAS, CHAQ, S-100 protein, CRP, hs-CRP, and ESR.(***iii***)
*
**The frequency of respiratory illness episodes and antibiotic administration at the end of the study were determined.**
*
(***iv***)***Any AE during the six-month follow-up period after the baseline:*** fever and injection site reactions.

### 2.5. Statistical Analysis

The sample size for a studied population with a margin of error of 5% and 95% significance is 189 patients. The sample size for an adverse reaction with a margin of error of 1% and 99% significance is 362 patients. All statistical calculations were performed in the R version 4.3.1 software package (R Foundation for Statistical Computing, Vienna, Austria).

For all numerical data, preliminary testing for the normality of the distribution was performed using the Shapiro–Wilk test. Most of the data did not follow the normal distribution. Quantitative data were presented as medians with interquartile ranges (IQRs) or geometric means with standard deviations; the number of missing values (N) was also calculated. For qualitative variables, we provide the absolute number (n), as well as the percentage (%) and 95% Confidence Interval (CI) for the percentages.

A logarithmic analysis of antibody levels was conducted before and after vaccination. Changes in antibody titer after vaccination in each group were analyzed using a Friedman ANOVA and Kendall’s test for three dependent variables and a Wilcoxon matched pairs test for two dependent numerical variables. A comparison of the proportion of participants who had a protective antibody titer present before and after vaccination was performed using McNemar’s test, adjusted for continuity, separately for each group. Analyses were not possible when one of the proportions equaled 100%. Bonferroni correction was applied to avoid multiple comparisons. The differences are statistically significant at a *p*-value < 0.05 and a *p*-value < 0.017 (0.05/3) after Bonferroni correction for intergroup comparison, with each result having been tested multiple times. All values of statistical significance are based on two-tailed tests. Predictors influenced the outcome—the presence of the protective titer of anti-SP IgG and anti-Hib IgG was assessed with univariate and multivariate regression analyses.

### 2.6. Ethics

Written consent was obtained according to the Declaration of Helsinki. The local Ethics Committee approved the protocol of the trial of the National Medical Research Center of Children’s Health (protocol number 15 from 2 October 2017).

## 3. Results

### 3.1. The Demography of the Patients

This study included 371 patients: 234 girls (63.0%) and 137 boys (37.0%). The median age at the time of vaccination was 10.9 years (4.2). The majority of the patients belong to RF-negative polyarticular (n = 164, 44.2%) and persistent oligoarticular JIA categories (n = 167, 45.0%), with a relatively high portion of the patients having uveitis (n = 67, 18%). Every third patient had a chronic or relapsed course of ENT diseases, such as chronic tonsillitis, adenoiditis, otitis, sinusitis, etc. Biological treatment was carried out in 217 (58.5%) patients, biological monotherapy in 102 patients (27.5%), and a combination with non-biologic DMARDs in 115 (31.0%) patients The majority of patients received TNF blockers (n = 202, 93.1%); 136 (36.7%) patients received methotrexate. Of the total patients, 98 (26.4%) had active arthritis and 67 (18.1%) had uveitis: 51 (76.1%) had uveitis in remission, 7 (10.4%) had low-active uveitis, and 9 (13.4%) had active uveitis at the baseline. Data are shown in Table 1.

### 3.2. Vaccine Seroconversion

The levels of the Ig G anti-SP and anti-Hib antibodies increased 3 weeks after vaccination. Anti-SP antibodies slightly decreased from D22 to the EOS, and anti-Hib antibodies remained increased during the whole study as well as in a part of the patients with a protective titer. Nearly half of the patients had two-fold-increased anti-SP (45%) and anti-Hib (56%) titer on Day 22 and to the end of the study (50% and 53%, respectively). Data are shown in Table 2 and Figure 2. When analyzing the dynamics of antibody levels in three subgroups of patients, (1) methotrexate, (2) bDMARDs, and (3) a combination of methotrexate and bDMARDs, no significant differences were observed in the initial antibody levels against SP and Hib. After three weeks, however, differences in antibody levels against SP became evident (*p* = 0.034). The lowest levels were observed in patients who received a combination of methotrexate and bDMARDs. It is noteworthy that in each of these groups, there was an approximately two-fold increase in antibody levels (*p* = 0.0000001). No significant differences in levels of antibodies were noted between the three groups of patients three weeks after vaccination of the anti-Hib vaccine. Nevertheless, all three groups exhibited a 2–3-fold increase in titer (*p* = 0.0000001), with the highest increase observed in the group receiving combined therapy methotrexate with bDMARDs.

### 3.3. Analysis of the Predictors of the Vaccination Outcome

To evaluate the predictors of reaching the protective titer of the antibodies to each vaccine, we used univariate and multivariate regression analyses. The level of the CHAQ was negative, but the ESR level had a positive impact on the protective anti-SP titer at the end of the study. Previous or current treatment with non-biologic and biologic DMARDs did not affect the protective anti-SP titer (Table 3).

Treatment with methotrexate and biologic drugs negatively impacted the protective titer of Hib antibodies at the end of the study (Table 4).

### 3.4. JIA Outcomes During Vaccination

During the study, there were no patients with significant flares, and the main JIA outcomes gradually decreased during the trial. The medians of all JIA outcomes, active joints, morning stiffness, MDVAS, pVAS, CHAQ, ESR, and CRP, decreased gradually during the study (Table 2). There was no significant rise in the hs-CRP or S100 protein after the vaccination. The number of patients with uveitis remained equal, as well as the part of the patients with active, low-active, and inactive uveitis.

### 3.5. The Frequency of Acute Infectious Events and the Use of Antibacterial Therapy

During the six months before vaccination, at least one episode of acute respiratory infection (ARI) was noted in nearly all study participants, and 70% of patients received antibiotics. Within 6 months following vaccination, the number of ARI episodes (*p* < 0.001) and the number of courses of antibacterial treatment (*p* < 0.0001) decreased twice. The median duration of ARI episodes decreased four times (*p* < 0.0001).

There were 46 patients who had previously had at least one episode of ARI and did not develop a single episode of ARI after vaccination, and there were 111 patients who had at least one course of antibacterial treatment previously and did not need a single course of antibacterial treatment after vaccination. The data are shown in Table 5.

### 3.6. Assessment of the Adverse Events

#### 3.6.1. Mild Adverse Events

Adverse events occurred in 58 (15.6%) patients with JIA. The most common vaccine-associated AE was swelling and/or tenderness at the injection site in 27 (7.3%) children. Fever (body temperature ≥ 38 °C) for 3 days following vaccination was reported in 10 (2.7%) patients. All episodes of fever resolved without any specific treatment within three days after vaccination. Swelling at the injection site was reported in 20 (5.4%) individuals and self-resolved within 2–3 days.

#### 3.6.2. Severe AE

One (0.27%) patient treated with etanercept and leflunomide being in remission on the third day after vaccination developed acute gastroenteritis, received antibiotics, and 17 days later developed macrophage activation syndrome (fever up to 39.5 °C, thrombocytopenia, hyperferritinemia, increased transaminases, lactate dehydrogenase, and C-reactive protein), which was controlled by intravenous methylprednisolone at a dose of 10 mg/kg per day, with a gradual decrease until complete withdrawal, in combination with antibiotic therapy.

## 4. Discussion

Our study has demonstrated the efficacy and safety of simultaneous vaccination against pneumococcus and Hib infection in children with JIA. The main benefits of this study are the decreased number and decreased severity of ARI episodes without JIA flares. Previous studies have shown the separate efficacy and safety of vaccination against pneumococcus and Hib infection in patients with rheumatic diseases on immunosuppressive treatment [11,21]. However, the efficacy and safety of simultaneous vaccination in children with JIA have not been previously studied. Infection prevention is essential in the management of patients with JIA [22]. Pneumococcal and Hib infections are two common diseases among immunocompromised patients, and the study has confirmed the safety and effectiveness of a simultaneous vaccination against these two infections [6,7]. Simultaneous vaccination with a 23-valent pneumococcal polysaccharide vaccine and a quadrivalent influenza vaccine showed a level of immunogenicity comparable to sequential vaccination without an increase in the number of adverse reactions [23]. During our study, the titer of antibodies against both pneumococcus and Hib in patients receiving different anti-rheumatic treatments increased. Neither biologics nor methotrexate affected the protective antibody titer against pneumococcal infection.

Biologics and methotrexate affected the protective titer of the antibodies against Hib, but the negative effect of the methotrexate was borderline. These findings are consistent with other studies [24,25]. A single case of JIA flare associated with acute gastroenteritis during vaccination occurred in our study. Twenty-five JIA patients were treated with methotrexate, etanercept, and adalimumab, and all achieved a satisfactory response on the 13-valent pneumococcal conjugate vaccine (PCV13) [25]. A previous study showed that methotrexate compared to anti-TNF therapy impacted the immunogenicity of the 23-valent polysaccharide pneumococcal vaccine in children with JIA [26].

Vaccination against pneumococcal and Hib infections in JIA patients does not increase biomarkers of JIA activity, such as hsCRP and calprotectin. The findings are consistent with the results of a study involving 125 children with JIA, who were treated either with methotrexate or in combination with etanercept, a TNF inhibitor, and who received vaccination against pneumococcus infection [25]. In adult rheumatoid arthritis patients, treatment with etanercept PCV13 vaccination was safe and effective [27].

In our study, we observed a rise in calprotectin and hsCRP concentrations and a decrease in CRP after the first three weeks after the vaccination, with a following declination in calprotectin and CRP, but hsCRP continued to rise. It should be noted that all increases in inflammatory biomarkers were statistically significant but were during the normal range and were not clinically meaningful due to the absence of clinical flares. The discrepancy between hsCRP and CRP is related to the different sensitivity of both tests in patients having normal CRP levels. The high-sensitivity CRP test is much more sensitive in the area of low–normal CRP levels. Walker et al. (2016) reported that the pneumococcus vaccine can cause severe local and systemic inflammatory reactions in patients with cryopyrin-associated syndromes (CAPS) and other autoinflammatory disorders [28]. Different types of PCV13 have similar safety profiles [29]. In our study, the vaccine-associated reactions accounted for 11% and were self-resolved. A review of 37 studies involving more than 2500 patients concluded that vaccination does not affect the activity of rheumatic diseases [30]. Another review of 138 JIA patients treated with methotrexate and glucocorticoids found that vaccinations for PCV and Hib did not lead to flares of the disease [31]. In this study, there was also no increase in calprotectin and hsCRP levels after 4 weeks of treatment.

The impact of vaccination against Streptococcus pneumoniae and *Haemophilus influenzae* type b on laboratory markers of disease activity, such as ESR and CRP, in patients with JIA, has not been investigated.

It has been reported that among patients with rheumatoid arthritis (n = 41) after 12 weeks of treatment with the live attenuated vaccine, CRP and ESR levels remained unchanged. By this time, approximately 88% of patients remained in remission, approximately 7% showed low disease activity (2.6 < DAS28 < 3.2), and approximately 5% showed moderate disease activity (3.2 < DAS28 ≤ 5.1). Approximately 6% were in remission. Of the patients with rheumatoid arthritis, 15% experienced an exacerbation of arthritis (ΔDAS28 > 1.1) within 6 to 12 weeks after vaccination. Among these patients, four had transient arthritis that resolved spontaneously or after treatment with low-dose glucocorticosteroids and two required therapy with a tumor necrosis factor-alpha (TNFα) inhibitor [32].

Data on antibody persistence after immunization with the 23-valent polysaccharide pneumococcal vaccine were presented in a similar study conducted in adults, albeit at different time points [33]. Our study also showed that the antibody level remained at a significant level for 6 months after vaccination. Treatment with methotrexate and biologic medications had a slightly negative impact on the protective level of Hib antibodies at the end of the study.

## 5. Limitations

The limitations of the study are related to the fact that some patients had received, and some had not received, vaccination against pneumococcus and/or Hib infection within the national immunization schedule before the onset of arthritis. Different time intervals from routine vaccination to study inclusion, the varying duration of previous anti-rheumatic treatment, the various doses of medications used in patients both before and during study participation, the possibility of multiple medications being used before study enrollment, and the diverse JIA categories were also limitations. The presence of a control group increased the significance of vaccination in reducing the frequency and duration of acute respiratory infection (ARI) episodes and the need for antibacterial drugs. The calculation of the number and duration of infectious episodes, as well as the need for antibiotic treatment, may have been affected by the retrospective nature of the study, the possibility that patients or their parents provided inaccurate data, or the omission of information from medical records. The season of the year, the current epidemiological situation, and the personal opinion of local physicians regarding the prescription of antibiotics may also influence the accuracy of the calculations. Additionally, we cannot determine whether co-administration of vaccines strengthens or weakens the immune response compared to single-vaccine administration.

## 6. Conclusions

Immunization with a 13-valent pneumococcal conjugate vaccine and an Hib vaccine in patients with JIA without systemic manifestations is an effective and safe tool for preventing infections. Seroconversion, the survival of vaccine-specific antibodies to pneumococcus and Hib infection for up to 6 months following vaccination, without a significant rise in clinical and laboratory markers of JIA activity (active joints and duration of morning stiffness), and predictors of disease flares (hsCRP and S-100 protein levels) were analyzed. Simultaneous vaccination against pneumococcal and Hib infections reduces the frequency and duration of episodes of acute respiratory infections, as well as the number of courses of antibiotics, and does not lead to JIA flares. The number of reported adverse events is consistent with what is expected.

## Figures and Tables

**Figure 1 vaccines-13-00177-f001:**
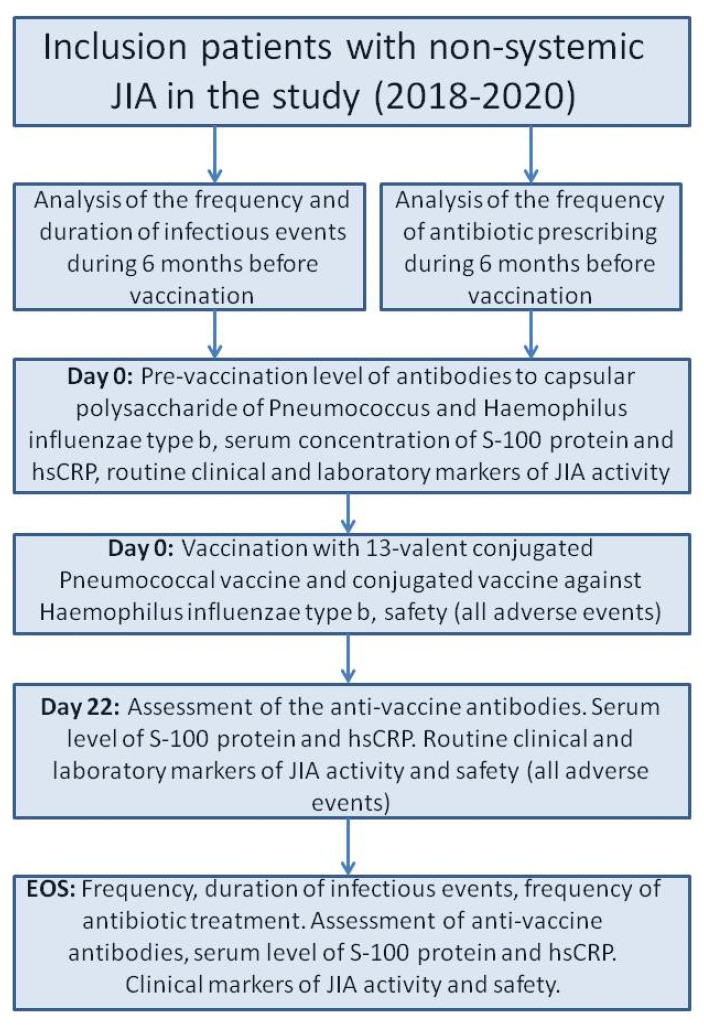
The flow chart of the study design. Abbreviations: EOS—end of study, hs-CRP—high-sensitivity CRP, JIA—juvenile idiopathic arthritis.

**Figure 2 vaccines-13-00177-f002:**
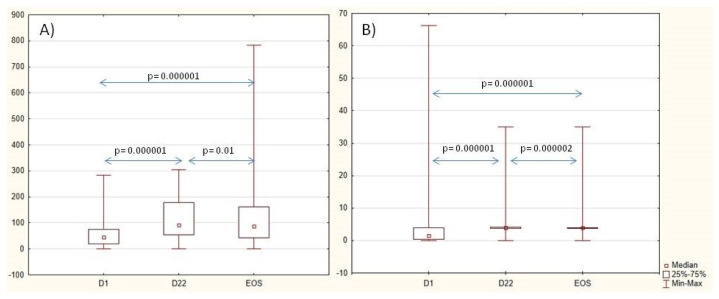
Dynamics of anti-SP IgG antibodies, U/mL (**A**), and anti-Hib antibodies, U/mL (**B**), during the study. *Abbreviations:* D—day, EOS—end of study.

**Table 1 vaccines-13-00177-t001:** Patients’ characteristics at baseline.

JIA Features	Results, n = 371 (%)
**Demography**	
Sex, female, n (%)	234 (63)
Age at baseline *, years ± SD	10.9 ± 4.2
JIA category, n (%)	
Extended oligoarthritis	2 (0.5)
Persistent oligoarthritis	167 (45)
Polyarthritis, RF-negative	164 (44.2)
Polyarthritis, RF-positive	12 (3.2)
Psoriatic arthritis	2 (0.5)
Enthesitis-related arthritis	24 (6.5)
Uveitis, n (%)	67 (18)
ENT diseases, n (%)	119 (32)
**Treatment at baseline**	
NSAIDs, n (%)	13 (3.5)
Oral corticosteroids, n (%)	3 (0.8)
Methotrexate, n (%)	136 (36.7)
Cyclosporine A, n (%)	27 (5.9)
Sulfasalazine, n (%)	8 (2.2)
Leflunomide, n (%)	7 (1.9)
Mofetyl mycophenolate, n (%)	1 (0.3)
**Biologic treatment ***	217 (58.5)
Etanercept	70 (18.9)
Adalimumab	26 (7.0)
Tocilizumab	6 (1.6)
Etanercept + nbDMARD	54 (14.0)
Adalimumab + nbDMARD	52 (14.0)
Tocilizumab + nbDMARD	9 (2.4)

*Abbreviations:* ENT—ear, nose, and throat; JIA—juvenile idiopathic arthritis; nbDMARD—non-biological disease-modifying anti-rheumatic drugs; NSAID—non-steroidal anti-inflammatory drugs; SD—standard deviation; * calculation made on whole JIA group (n = 371).

**Table 2 vaccines-13-00177-t002:** Dynamics of anti-SP and anti-Hib antibodies and the main JIA-associated outcomes during the study.

Parameters (n = 371) n (%)/Me (IQR) [Min; Max]	D1	D22	EOS	p_1_	p_2_	p_3_	*p* Total
Anti-SP IgG titer, U/mL *	44 (19; 74) [0–283]	90 (52; 179) [0–304]	83 (40; 162) [0–784]	0.000001	0.000001	0.01	0.000001
Protective titer anti-SP IgG, n (%) 95% CI	339 (91) [88; 94]	359 (97) [95; 98]	351 (95) [92; 97]	<0.001	<0.001	>0.05	<0.001
Patients with a two-fold-increased anti-SP titer, n (%) [95% CI]	-	168 (45) [40; 51]	185 (50) [45; 52]	-	<0.001	<0.001	<0.001
Anti-Hib titer, U/mL, median (IQR) [min; max]	1.2 (0.3; 4.0) [0.0; 66.3]	4.0 (4.0; 4.0) [0; 35]	4.0 (3.8; 4.0) [0.1; 35.0]	0.000001	0.000001	0.00002	0.000001
Patients with a protective anti-HIb titer, n (%) [95% CI]	200 (54) [49; 59]	355 (96) [93; 97]	346 (94) [90; 97]	<0.001	<0.001	<0.001	<0.001
Patients with a two-fold-increased anti-HIb titer, n (%) [95% CI]	-	209 (56) [51; 61]	196 (53) [48; 58]	-	<0.001	>0.05	<0.001
S-100 U/mL (n.v. > 2.9), Me (IQR) [min; max]	2.3 (1.3; 3.9) [0.0; 95.0]	2.6 (1.4; 4.4) [0.0; 169.0]	0.67 (0.3; 1.7) [0.0; 168.0]	0.024	0.001	0.001	0.000001
hs-CRP, U/mL (n.v. > 8.2) Me (IQR) [min; max]	0.6 (0.2; 1.6) [0.0; 61.5]	0.7 (0.3; 1.9) [0.0; 69.0]	0.78 (0.3; 2.0) [0.0; 41.3]	0.001	0.179	0.182	0.001
Active joints, Me (IQR) [min; max]	0.0 (0.0; 1.0) [0.0; 44.0]	0.0 (0.0;0.25) [0.0; 36.0]	0.0 (0.0; 0.0) [0.0; 28.0]	0.002	0.000001	0.000001	0.000001
Joints with limited ROM, Me (IQR) [min; max]	0.0 (0.0; 2.0) [0.0; 45.0]	0.0 (0.0; 2.0) [0.0; 45.0]	0.0 (0.0; 0.0) [0.0; 28.0]	0.009	0.000001	0.000006	0.000001
Morning stiffness, min, Me (IQR) [min; max]	0 (0; 0) [0; 360]	0 (0; 6) [0; 180]	0 (0; 0) [0; 180]	0.000003	0.000001	0.000001	0.000001
MDVAS, mm, Me (IQR) [min; max]	0 (0; 8) [0; 88]	0 (0; 10) [0; 85]	0 (0; 0) [0; 56]	0.000003	0.0000001	0.0000001	0.000001
Patient/parent VAS, mm, Me (IQR) [min; max]	0 (0; 10) [0; 90]	0 (0; 10) [0; 88]	0 (0; 0) [0; 66]	0.000001	0.0000001	0.0000001	0.000001
CHAQ, points, Me (IQR) [min; max]	0.0 (0.0;0.13) [0.0; 2.8]	0.0 (0.0; 0.0) [0.0; 2.5]	0.0 (0.0; 0.0) [0.0; 2.3]	0.01	0.0000001	0.000001	0.000001
ESR, mm/hr (n.v = 0–20) Me (IQR) [min; max]	3 (2; 8) [0; 67]	4 (2; 6) [0; 50]	3 (2; 6) [0; 53]	0.003	0.026	0.473	0.016
CRP, mg/mL (n.v. < 5) Me (IQR) [min; max]	1.0 (0.5; 1.2) [0.0; 74.1]	1.0 (0.4; 1.1) [0.0; 105.1]	0.7 (0.3; 1.3) [0.0; 40.2]	0.024	0.0002	0.01	0.000001

Footnotes: p_1_—comparison between D1 and D22; p_2_—comparison between D1 and EOS; p_3_—comparison between D22 and EOS; * Me (IQR) [min; max]. Abbreviations: CHAQ—Childhood Health Assessment Questionnaire; CRP—C-reactive protein; Hib—*Haemophilus influenza* type b; hsCRP—high-sensitivity CRP; ESR—erythrocyte sedimentation rate; Me—median; IQR—interquartile range; ROM—range of motion; VAS—visual analogue scale; MDVAS—physician’s VAS; SP—*S. pneumoniae*.

**Table 3 vaccines-13-00177-t003:** Predictors of the protective titer of anti-SP IgG to the end of the study.

Characteristic	Univariate		Multivariate	
	OR (95% CI)	*p*-Value	OR (95% CI)	*p*-Value
bDMARD before vaccination				
did not receive bDMARD	—		—	
received bDMARD	0.75 (0.27: 1.87)	0.5	0.73 (0.23; 2.42)	0.6
DMARD before vaccination				
did not receive methotrexate	—		—	
methotrexate	1.15 (0.42; 2.89)	0.8	0.96 (0.30; 3.17)	>0.9
S-100 before vaccination	0.98 (0.95; 1.04)	0.4	0.98 (0.94; 1.03)	0.2
CRP before vaccination	1.03 (0.96; 1.21)	0.6	0.97 (0.90; 1.11)	0.5
Active joints	1.02 (0.93; 1.26)	0.8	1.19 (0.96; 1.61)	0.2
CHAQ	0.68 (0.31; 1.97)	0.4	0.18 (0.04; 0.92)	0.026
ESR	1.16 (1.02; 1.43)	0.072	1.21 (1.05; 1.50)	0.041

*Abbreviations:* CHAQ—Childhood Health Assessment Questionnaire; CI—Confidence Interval; CRP—C-reactive protein; ESR—erythrocyte sedimentation rate; OR—Odds Ratio; SP—*S. pneumoniae.*

**Table 4 vaccines-13-00177-t004:** Predictors of the protective titer of anti-Hib IgG at the end of the study.

Characteristic	Univariate		Multivariate	
	OR (95% CI)	*p*-Value	OR (95% CI)	*p*-Value
bDMARD before vaccination				
did not receive bDMARD	—		—	
received bDMARD	0.45 (0.16; 1.11)	0.10	0.29 (0.10; 0.79)	0.019
DMARD before vaccination				
did not receive methotrexate	—		—	
methotrexate	0.69 (0.24; 1.69)	0.4	0.39 (0.13; 1.05)	0.072
S-100 before vaccination	1.11 (0.98; 1.37)	0.2	1.08 (0.98; 1.33)	0.3
CRP before vaccination	1.02 (0.96; 1.15)	0.7	1.02 (0.95; 1.16)	0.6
Active joints	0.97 (0.91; 1.06)	0.3	0.91 (0.75; 1.06)	0.2
CHAQ	0.89 (0.39; 2.81)	0.8	2.34 (0.46; 25.5)	0.4
ESR	1.0 (0.96; 1.07)	0.9	1.0 (0.95; 1.07)	0.9

*Abbreviations:* CHAQ—Childhood Health Assessment Questionnaire; CI—Confidence Interval; CRP—C-reactive protein; ESR—erythrocyte sedimentation rate; Hib—*Haemophilus influenza* type b; OR—Odds Ratio.

**Table 5 vaccines-13-00177-t005:** The dynamics of acute respiratory episodes before combined vaccination and after.

Infection Burden Indicators, Me (IQR) [Min; Max]	6 Months Prior to Vaccination	6 Months After Vaccination	*p*-Value
Duration of ARI episode *, days	9.0 (5.0; 10.0)	2.0 (1.0; 3.0)	<0.001
	[0.0; 19.0]	[0.0; 15.0]	
The number of ARI episodes per patient *	4.0 (3.0; 5.0)	2.0 (1.0; 2.0)	<0.001
	[0.00; 9.00]	[0.0; 5.0]	
The number of courses of antibacterial drugs *	2.0 (2.0; 3.0)	1.0 (0.0; 1.0)	<0.001
	[0.0; 6.0]	[0.0; 3.0]	
Patients without ARI, n (%)	21 (5.7)	62 (16.7)	<0.001
Patients did not require antibacterial drugs, n (%)	40 (10.8)	87 (23.5)	<0.001

*Abbreviations:* ARI—acute respiratory infection; * calculation was conducted during 6-month period before and after the vaccination.

## Data Availability

Data are available on request from the authors.

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
