# Peer review of "The Efficacy and Safety of Simultaneous Vaccination with Polysaccharide Conjugate Vaccines Against Pneumococcal (13-Valent Vaccine) and Haemophilus influenzae Type b Infections in Children with Juvenile Idiopathic Arthritis Without Systemic Manifestations: A Prospective Cohort Study"

_vaccines, 2025, doi:10.3390/vaccines13020177_

Round 1

Reviewer 1 Report

Comments and Suggestions for Authors

In this paper the authors have analyzed prospectively  the effects of the simultaneous vaccination with a 13-valent vaccine against pneumococcal and Haemophilus influenza type B in 371 children with juvenile idiopathic arthritis; they observed a significant increase of antibodies against both pathogens and a decrease in the frequency and duration of the episodes of acute respiratory infections.

Major comments:

1) In general, an effort should be made in writing, abbreviations are used and defined only afterwards, abbreviations are repeated (JIA line 314, anti-SP titer line 156, and the abbreviation indicated line 194, capital letters are missing for some bacteria,....)

2) The sampling techniques need to be clarified: the authors talk about sampling on EDTA (line 150) and then talk about serum. It is rather plasma if the blood was collected on EDTA.

3) Why the sample size was not calculated initially ? (line 205).

4) In the material & methods section, the authors indicate they quantified hsCRP and CRP (results in table 2). However, there is a significant increase of hsCRP between D1 and D22 (p = 0.001), whereas CRP decreased significantly (p = 0.024). CRP are in the normal range, and this decrease is certainly due to the low sensitivity of the test when levels are low; Consequently, why the authors quantified both markers ? More importantly, the authors write in the discussion that there was no increase in hsCRP after 4 weeks of treatment (line 347). This is not supported by the levels indicated in the table, and hsCRP was quantified at D22 and not after 4 weeks. All the variatons of CRP and hsCRP should be adequatly discussed.

5) Line 332: the authors indicate that a study in China observed lower titers of antibodies in children with JIA. tHe authors should discuss about the different results between both studies.

Minor comments:

1) lines 140to 142: problem of syntax. "We utilized the kits......were used, respectively.

2)The flow chart: capital are missing for Pneococcus and Hemophilus.,, a dot is missing between "events" and "Frequency"

3) line 172: A acpital letter is missing "all"

Author Response

Reviewer 1.

In this paper the authors have analyzed prospectively  the effects of the simultaneous vaccination with a 13-valent vaccine against pneumococcal and Haemophilus influenza type B in 371 children with juvenile idiopathic arthritis; they observed a significant increase of antibodies against both pathogens and a decrease in the frequency and duration of the episodes of acute respiratory infections.

Dear Reviewer!

Thank you for your kind evaluation of our manuscript. Our answers (A) on your queries (Q) are below. All changes in the manuscript highlighted with color.

Major comments:

Q1) In general, an effort should be made in writing, abbreviations are used and defined only afterwards, abbreviations are repeated (JIA line 314, anti-SP titer line 156, and the abbreviation indicated line 194, capital letters are missing for some bacteria,....)

A1) Dear Reviewer! All recommended changes were done.

Q2) The sampling techniques need to be clarified: the authors talk about sampling on EDTA (line 150) and then talk about serum. It is rather plasma if the blood was collected on EDTA.

 A2) Dear Reviewer! Of course, plasma. The serum was substitute to plasma everywhere.

Q3) Why the sample size was not calculated initially? (line 205).

A3) Dear Reviewer! Sorry for the error. It was calculated initially. This sentence has deleted.

Q4) In the material & methods section, the authors indicate they quantified hsCRP and CRP (results in table 2). However, there is a significant increase of hsCRP between D1 and D22 (p = 0.001), whereas CRP decreased significantly (p = 0.024). CRP are in the normal range, and this decrease is certainly due to the low sensitivity of the test when levels are low; Consequently, why the authors quantified both markers ? More importantly, the authors write in the discussion that there was no increase in hsCRP after 4 weeks of treatment (line 347). This is not supported by the levels indicated in the table, and hsCRP was quantified at D22 and not after 4 weeks. All the variatons of CRP and hsCRP should be adequatly discussed.

A4) Dear Reviewer! When the study was planned our idea was to use some markers, which might interfere the risk of flare, especially on the preclinical stage. So we used S100 protein and hsCRP, because in literature they were positioned as possible predictors of the flare. During the study we observed the significant raise of hsCRP level, but this elevation was during the normal range. We did not see the clinical flares. Interesting, the usual CRP, significantly decreased, but this down was also during the normal range without clinical meanings. The summary of this point: we did not observed preclinical flare risks, due to absence of the clinical meaningful flares. The main discrepancy between these tests are related with different sensitivity in the area of low CRP levels, usually less than 5 mg/l and especially less than 1 mg/l. The part of the discussion with the error was fixed now. Thank you again

Q5) Line 332: the authors indicate that a study in China observed lower titers of antibodies in children with JIA. He authors should discuss about the different results between both studies.

 A5) Dear Reviewer! The study in China had the control group, the Authors compared the results in JIA cohort and controls and JIA patients had lower titers of anti-vaccine antibodies, which is relevant to the majority of the similar anti-vaccine antibodies studies in pediatric rheumatic disease patients. Because this sentence is far from the results of our study I decided to delete this sentence and the reference

Minor comments:

Q1) lines 140to 142: problem of syntax. "We utilized the kits......were used, respectively.

A1) Dear Reviewer! This sentence has fixed now. The new sentence is “We employed commercial test kits to assess the concentration of high-sensitivity C-reactive protein (hsCRP) (Biomerica, Germany), employing a solid-phase indirect enzyme immunoassay and kits for measuring the levels of MRP8/14 (S100A8/A9, calprotectin) (BÜHLMANN, Switzerland) for conducting a sandwich ELISA”.

Q2)The flow chart: capital are missing for Pneococcus and Hemophilus.,, a dot is missing between "events" and "Frequency"

A2) Dear Reviewer! All you recommendations done.

Q3) line 172: A acpital letter is missing "all"

A3) Dear Reviewer! Fixed. Thank you!

Dear Reviewer!

I hope manuscript has become better after your suggestions.

Thank you so much!

On behalf of the Authors,

Mikhail Kostik, MD, Ph.D., Professor

Reviewer 2 Report

Comments and Suggestions for Authors

The authors have studied the efficacy and safety of simultaneous vaccination with Pneumococcal vaccine and Hemophilus influenza type b vaccine in 371 pediatric patients with juvenile idiopathic arthritis. This was a prospective study, where the patients were followed totally 6 months after vaccination. The number of respiratory infections during the follow-up was compared to the 6 months before vaccination. However, it is unclear when did the patients enter the actual study and how the data on infections prior to vaccination was collected.

Furthermore, there was no control groups, which make it difficult to evaluate the impact of seasonal variation of infections on the study results.

The authors also report the antibody titres up to 3 weeks after vaccination showing a good response. There was no analysis of subgroups receiving different anti-rheumatic drugs.

The authors also studied the safety of the vaccination. There was, however, no sample size calculation for the power of detecting rare adverse reactions.

The presentation and language (including the topic "Hemophilic Type b Infections...") should be improved before publication of the manuscript. 

Comments on the Quality of English Language

The presentation and language (including the topic "Hemophlic Type b infections...) should be improved before publication of the manuscript. 

Author Response

Reviewer 2.

The authors have studied the efficacy and safety of simultaneous vaccination with Pneumococcal vaccine and Hemophilus influenza type b vaccine in 371 pediatric patients with juvenile idiopathic arthritis. This was a prospective study, where the patients were followed totally 6 months after vaccination. The number of respiratory infections during the follow-up was compared to the 6 months before vaccination.

Dear Reviewer! Thank you for your kind evaluation of our manuscript. Our answers (A) on your queries (Q) are below. All changes in the manuscript highlighted with color.

Q1) However, it is unclear when did the patients enter the actual study and how the data on infections prior to vaccination was collected.

A1) Dear Reviewer! All patients were included in 2018-2020. The information is in the Methods section and in flow-chart. All information about infection prior to vaccination was collected from the patients’ medical documents and after the baseline by asking the patients and parents, and analyzing the medical charts of the patients. The information added in the relevant part of the Methods section

Q2) Furthermore, there was no control groups, which make it difficult to evaluate the impact of seasonal variation of infections on the study results.

A2) Dear Reviewer! This is the weak part of this study. We originally didn't plan a control group, and now it's hard to find a control group of the same years. This weak part of our study is noted in the Limitations section 

Q3) The authors also report the antibody titres up to 3 weeks after vaccination showing a good response. There was no analysis of subgroups receiving different anti-rheumatic drugs.

A3) Dear Reviewer! We did the additional analysis in three subgroups: 1) Methotrexate; 2) Biologics;    3) Methotrexate with Biologics. At the baseline there were no significant differences between Sp titer between these three groups, but the level in the third group was lower compared groups 1 and 2. In every group during the first three weeks the anti-Sp level increase twice (p=0.0000001 for every group), but the significant difference was observed: patients with combined treatment had lower anti-Sp titers compared Methotrexate or Biologics (p=0.034). There were no differences between the groups at baseline and D22. The initial antiHib level was higher in Methotrexate group, but the data are nonsignificant.  In every group the titer raise at least 2-3 times. Interestingly, the highest raise (3 times) was oserved in Methotrexate+Biologics groups, having initially the lowest titer of antiHib antibodies. The information added in the manuscript

Q4) The authors also studied the safety of the vaccination. There was, however, no sample size calculation for the power of detecting rare adverse reactions.

A4) Dear Reviewer! The sample size for adverse reaction with error of 1% and 99% significance is 362 patients. The information added in the statistics section.

Q5) The presentation and language (including the topic "Hemophilic Type b Infections...") should be improved before publication of the manuscript. 

A 5) Dear Reviewer! The language corrections were made.

Dear Reviewer!

I hope manuscript has become better after your suggestions.

Thank you so much!

On behalf of the Authors,

Mikhail Kostik, MD, Ph.D., Professor

Reviewer 3 Report

Comments and Suggestions for Authors

Dear Authors,

After reading the manuscript entitled “The Efficacy and Safety of Simultaneous Immunization with Polysaccharide Conjugate Vaccines Against Pneumococcal (13-Valent Vaccine) and Hemophilic Type b Infections in Children with Juvenile Idiopathic Arthritis Without Systemic Manifestations: A Prospective Cohort Study” I submit here my comments for improvements: 

General considerations:

The manuscript describes interesting finding on the usefulness of combined vaccination (pneumococcal -13-valent vaccine, Pfizer- and Haemophilus influenzae type B, Hib-,GSK) on a population of JIA patients. 

The manuscript visual appeal to the reader is very poor, since there is no graphs or colored figures backing up the tables and numbers. I suggest to have at least the vaccine specific antibody responses displayed in a self explanatory graph, and if possible do the same for other indicators.

Specific issues:

1) In the title, please replace Immunization by the more correct term "Vaccination" and correct in the title and elsewhere the mistaken term "Hemophilic" the correct name is : Haemophilus influenzae type b (Hib) -to not be confused with Haemophilia, a genetic disorder-. 

2)In line 24, please introduce the name before each acronym: what is SP? 

3) In line 35 introduce first the meaning before using the acronym: what is DMARDS? 

4) In line 66 change Haemophilic as sugested.

5) In line 77. ad the word “Moreover” to express “ Moreover, vaccination coverage...”

6) In line 79 change the phrase for : "may decrease the protective immunity generated by vaccines".

7) In Line 88 change Haemophilic as suggested.

8) In Line 122 Correct to Haemophilus influenzae and italicize it (species names must be in Italic).

9) In Line 278 there are word in a different type size, please correct.

10) From line288 to line 291, please replace the whole sentence by: "There were 46 patients who had previously at least one episode of ARI and did not develop a single episode of ARI after vaccination, and 111 patients who had at least one course of antibacterial treatment previously, and did not need a single course of antibacterial treatment after vaccination.”

11) In line 322 correct 23-valent to 13-valent.

12) About what you wrote on lines 322 to 325, please reformulate to make sense of the previous data, in the context of concomitant administration of two vaccines against bacterial diseases.

13) On lines 327 to 329 please reformulate to: "...neither biologics nor methotrexate affected the protective titer of the antibodies against pneumococcus infection. "

14) Line 370: change 23-valent to 13-valent.

15) Line 390: please change “anti-vaccine antibodies” here and elsewhere to "vaccine specific antibodies" or "vaccine humoral response".

Based on all the aforementioned, now I recommend you to do all these minor changes and resubmit the corrected version for publication. If all the suggested changes are included, the manuscript may be in shape to be published in Vaccines

My best regards,

The Reviewer.

Author Response

Reviewer 3.

Dear Authors,

After reading the manuscript entitled “The Efficacy and Safety of Simultaneous Immunization with Polysaccharide Conjugate Vaccines Against Pneumococcal (13-Valent Vaccine) and Hemophilic Type b Infections in Children with Juvenile Idiopathic Arthritis Without Systemic Manifestations: A Prospective Cohort Study” I submit here my comments for improvements: 

Dear Reviewer! Thank you for your kind evaluation of our manuscript. Our answers (A) on your queries (Q) are below. All changes in the manuscript highlighted with color.

General considerations:

Q1) The manuscript describes interesting finding on the usefulness of combined vaccination (pneumococcal -13-valent vaccine, Pfizer- and Haemophilus influenzae type B, Hib-,GSK) on a population of JIA patients. The manuscript visual appeal to the reader is very poor, since there is no graphs or colored figures backing up the tables and numbers. I suggest to have at least the vaccine specific antibody responses displayed in a self explanatory graph, and if possible do the same for other indicators.

A1) Dear Reviewer! The figure has been added.

Specific issues:

 Q1) In the title, please replace Immunization by the more correct term "Vaccination" and correct in the title and elsewhere the mistaken term "Hemophilic" the correct name is : Haemophilus influenzae type b (Hib) -to not be confused with Haemophilia, a genetic disorder-. 

A1) Dear Reviewer! Fixed.

Q2) In line 24, please introduce the name before each acronym: what is SP? 

A2) Dear Reviewer! SP- S.pneumoniae. The name added before the acronym.

Q3) In line 35 introduce first the meaning before using the acronym: what is DMARDS? 

A3) Dear Reviewer! Done.

Q4) In line 66 change Haemophilic as sugested.

 A4) Dear Reveiwer! Done.

Q5) In line 77. ad the word “Moreover” to express “ Moreover, vaccination coverage...”

A5) Dear Reviewer! Thank you for your suggestion. Added.

Q6) In line 79 change the phrase for : "may decrease the protective immunity generated by vaccines".

A6) Dear Reviewer! Done.

Q7) In Line 88 change Haemophilic as suggested.

 A7) Dear Reviewer! Fixed.

Q8) In Line 122 Correct to Haemophilus influenzae and italicize it (species names must be in Italic).

 A8) Dear Reviewer. Done and italicized everywhere.

Q9) In Line 278 there are word in a different type size, please correct.

A9) Dear Reviewer! Corrected.

Q10) From line288 to line 291, please replace the whole sentence by: "There were 46 patients who had previously at least one episode of ARI and did not develop a single episode of ARI after vaccination, and 111 patients who had at least one course of antibacterial treatment previously, and did not need a single course of antibacterial treatment after vaccination.”

A10) Dear Reviewer! Thank you so much for your help and suggestion. The sentence replaced.

Q11) In line 322 correct 23-valent to 13-valent.

 A11) Dear Reviewer! 23-valent is correct.

The title of the cited manuscript is “Nakashima K, Aoshima M, Ohfuji S, Yamawaki S, Nemoto M, Hasegawa S, Noma S, Misawa M, Hosokawa N, Yaegashi M, Otsuka Y. Immunogenicity of simultaneous versus sequential administration of a 23-valent pneumococcal polysaccharide vaccine and a quadrivalent influenza vaccine in older individuals: A randomized, open-label, non-inferiority trial. Hum Vaccin Immunother. 2018;14(8):1923-1930. doi: 10.1080/21645515.2018.1455476. Epub 2018 May 14. PMID: 29561248; PMCID: PMC6150043.”

Q12) About what you wrote on lines 322 to 325, please reformulate to make sense of the previous data, in the context of concomitant administration of two vaccines against bacterial diseases.

A12) Dear Reviewer! Done.

Q13) On lines 327 to 329 please reformulate to: "...neither biologics nor methotrexate affected the protective titer of the antibodies against pneumococcus infection. "

A13) Dear Reviewer! Fixed the errors.

Q14) Line 370: change 23-valent to 13-valent.

 A14) Dear Reviewer! I check the reference [33]. 23-valent is correct.

Q15) Line 390: please change “anti-vaccine antibodies” here and elsewhere to "vaccine specific antibodies" or "vaccine humoral response".

A15) Dear Reviewer! I used the term vaccine-specific antibodies during the text. thank you for you suggestions.

Based on all the aforementioned, now I recommend you to do all these minor changes and resubmit the corrected version for publication. If all the suggested changes are included, the manuscript may be in shape to be published in Vaccines

My best regards,

The Reviewer.

Dear Reviewer!

I hope manuscript has become better after your suggestions.

Thank you so much!

On behalf of the Authors,

Mikhail Kostik, MD, Ph.D., Professor

Round 2

Reviewer 2 Report

Comments and Suggestions for Authors

The authors state that this was a prospective cohort study. However, all information about infection prior to vaccination was collected retrospectively from the patients’ medical documents and after the baseline by asking the patients and parents, and analyzing the medical charts of the patients. The data on infections after vaccination, on the other hand, was collected on the basis of active follow-up and medical checks. The comparison of the number of infections before and after the vaccination is not done with the same precision and is therefore to some extent unreliable. Also the epidemiological situation in the area affects the number of seasonal infections in children (and not only the vaccinations). I am not convinced that the claim on the difference in the number of infections before and after vaccination is quite comparable. This should be stated in the discussion and limitations of the study.

Otherwise, the authors have responded adequately to the comments of the reviewers.

Comments on the Quality of English Language

There are still some odd wordings in the manuscript (including the word "hemophilic" in the topic).

Author Response

Reviewer 2.

Dear Reviewer!

Thank you for your kind evaluation of our manuscript. Our answers (A) on your queries (Q) are below. All changes in the manuscript highlighted with color.

Q1) The authors state that this was a prospective cohort study. However, all information about infection prior to vaccination was collected retrospectively from the patients’ medical documents and after the baseline by asking the patients and parents, and analyzing the medical charts of the patients. The data on infections after vaccination, on the other hand, was collected on the basis of active follow-up and medical checks. The comparison of the number of infections before and after the vaccination is not done with the same precision and is therefore to some extent unreliable. Also the epidemiological situation in the area affects the number of seasonal infections in children (and not only the vaccinations). I am not convinced that the claim on the difference in the number of infections before and after vaccination is quite comparable. This should be stated in the discussion and limitations of the study.

Otherwise, the authors have responded adequately to the comments of the reviewers.

A1) Dear Reviewer! We absolutely agree with your comments. The information about retrospective parts of the study added in the Methods. It was related to infectious episodes, antibacterial treatment and adverse events.

According to the possible inaccuracy in the assessment of the dynamics of the infectious episodes the additional information added in the Limitations section. “The calculation of the number and duration of infectious episodes, as well as the need for antibiotic treatment, may have been affected by retrospective nature of the study, the possibility that patients or their parents provided inaccurate data, or the omission of information from medical records. The season of the year, the current epidemiological situation, and the personal opinion of local physicians regarding the prescription of antibacterial drugs may also influence the accuracy of the calculations”. Based on the seasons and epidemiological situation, we included patients in our study over a three-year period during different seasons, which may have reduced the level of inaccuracy. All changes in the manuscript highlighted with color.

Q2) Comments on the Quality of English Language.

There are still some odd wordings in the manuscript (including the word "hemophilic" in the topic).

A2) Dear Reviewer! We have improved the quality of the English language and changed hemophilic on Haemophilus

Dear Reviewer!

I think the manuscript has become better after your suggestions.

Thank you so much!

On behalf of the Authors,

Mikhail Kostik, MD, Ph.D., Professor